# The mwtab Python Library for RESTful Access and Enhanced Quality Control, Deposition, and Curation of the Metabolomics Workbench Data Repository

**DOI:** 10.3390/metabo11030163

**Published:** 2021-03-12

**Authors:** Christian D. Powell, Hunter N.B. Moseley

**Affiliations:** 1Department of Computer Science (Data Science Program), University of Kentucky, Lexington, KY 40506, USA; christian.powell@uky.edu; 2Markey Cancer Center, University of Kentucky, Lexington, KY 40506, USA; 3Superfund Research Center, University of Kentucky, Lexington, KY 40506, USA; 4Department of Molecular and Cellular Biochemistry, University of Kentucky, Lexington, KY 40506, USA; 5Institute for Biomedical Informatics, University of Kentucky, Lexington, KY 40506, USA

**Keywords:** metabolomics workbench, data validation, data deposition, python package

## Abstract

The Metabolomics Workbench (MW) is a public scientific data repository consisting of experimental data and metadata from metabolomics studies collected with mass spectroscopy (MS) and nuclear magnetic resonance (NMR) analyses. MW has been constantly evolving; updating its ‘mwTab’ text file format, adding a JavaScript Object Notation (JSON) file format, implementing a REpresentational State Transfer (REST) interface, and nearly quadrupling the number of datasets hosted on the repository within the last three years. In order to keep up with the quickly evolving state of the MW repository, the ‘mwtab’ Python library and package have been continuously updated to mirror the changes in the ‘mwTab’ and JSONized formats and contain many new enhancements including methods for interacting with the MW REST interface, enhanced format validation features, and advanced features for parsing and searching for specific metabolite data and metadata. We used the enhanced format validation features to evaluate all available datasets in MW to facilitate improved curation and FAIRness of the repository. The ‘mwtab’ Python package is now officially released as version 1.0.1 and is freely available on GitHub and the Python Package Index (PyPI) under a Clear Berkeley Software Distribution (BSD) license with documentation available on ReadTheDocs.

## 1. Introduction

The Metabolomics Workbench (MW) is a public scientific data repository of metabolomics experimental datasets that was established in 2013 [1]. The repository consists of experimental data and metadata from metabolomics studies collected with mass spectroscopy (MS) and nuclear magnetic resonance (NMR) technologies. Specific projects, studies, and experiments (analyses) can be accessed via MW in ‘mwTab’ (text-based files) or JavaScript Object Notation (JSON) formatted files [2,3]. MW offers a web-based interface to analyze, track, deposit, or download data. Additionally, MW offers a REpresentational State Transfer (REST) interface to download and view data [4]. When the ‘mwtab’ Python package was first published in August 2017 (30 August 2017), the MW contained a total of 634 analyses. Over the last three years, the MW repository has continued to quickly grow and now contains more than 1000 projects as of November 2020, which can be subdivided into more than 1500 studies and more than 2500 individual analyses. The MW has maintained its ‘mwTab’ file format, consisting of text-based blocks, which use tabs to organize data, and published an updated specification guide at the beginning of 2019.

The ‘mwtab’ Python package was originally released in 2017 with the intent of creating a programmatic means of accessing and analyzing ‘mwTab’ formatted files from MW [5]. This package had the basic facilities for creating ‘mwTab’ formatted files to facilitate more automated deposition processes. In doing so, the package aimed to follow the FAIR data principles of findability, accessibility, interoperability, and reusability [6,7]. Since then the ‘mwtab’ package has seen a few updates, which have included various bug fixes and improvements. With the 1.0.1 release written in Python 3, we aimed to dramatically increase the functionality of the ‘mwtab’ package, while also updating the present methods to keep in line with changes in MW [8]. Release 1.0.1 includes updates to reflect the evolving changes in the ‘mwTab’ format specification, incorporates methods that utilize the MW’s REST interface, and provides a range of new validation functionality for quality control and curation purposes. These changes lay the foundations for expanding the ‘mwtab’ package functionality as a Command-Line Interface (CLI) and Python Application Programming Interface (API) to the MW data repository.

## 2. Results

### 2.1. Additional Functionality of the ‘mwtab’ Package Interface

The package can be used in two ways: (1) as an API within Python scripts (see Table 1) and (2) as a CLI (see Table 2). The 1.0.1 release of the ‘mwtab’ package includes a number of new features and changes for both API and CLI usage. The 1.0.1 release is now available for import via pip from the Python Package Index (PyPI) or can be manually installed from the GitHub repository [9,10].

### 2.2. Evaluation of the Metabolomics Workbench Repository

#### 2.2.1. Analysis IDs with Files Missing from the Metabolomics Workbench

As of 19 November 2020, a total of 1891 analyses were available for download through MW’s REST interface. When we attempted to download all available analyses, a number of analyses were not present for download in a given format. Of those analyses, three (3) could not be downloaded in ‘mwTab’ format and fifty (50) could not be downloaded in JSON format (see Table 3). Only blank pages were present for these files.

#### 2.2.2. Analysis Files Which Could Not Be Parsed

Of the 1888 downloaded analyses in ‘mwTab’ format, seventy (70) files could not be parsed into ‘~mwtab.mwtab.MWTabFile’ objects. Of the 1841 downloaded analyses in JSON format, 139 files could not be parsed into ‘~mwtab.mwtab.MWTabFile’ objects.

Of the seventy (70) ‘mwTab’ files which could not be parsed; fifty (50) files contained formatting errors in one or more data lines in their ‘SUBJECT_SAMPLE_FACTORS’ section, seven (7) files contained ‘MS_METABOLITE_DATA’ sections missing their ‘#’ prefix, four (4) files contained errors in the ‘VERSION’ line of their ‘METABOLOMICS WORKBENCH’ section, four (4) files contained a duplication of the ‘PR:INSTITUTE’ item in their ‘PROJECT’ section, two (2) files contained formatting errors in the ‘AN:ANALYSIS_TYPE’ line of their ‘ANALYSIS’ section, one (1) file was missing the tab delineator in the ‘SP:SAMPLEPREP_PROTOCOL_FILENAME’ line of its’ ‘SAMPLEPREP’ section, one (1) file contained a large number of excessive tabs in its ‘METABOLOMICS WORKBENCH’ header line, and one (1) file was missing the ‘METABOLOMICS WORKBENCH’ section in its entirety.

The 139 JSON files, which could not be parsed into ‘~mwtab.mwtab.MWTabFile’ objects, could not be parsed due to various formatting errors which broke the JSON format standard and prevented the loading of the file string into dictionary objects with the exception of a single file which lacked the ‘METABOLOMICS WORKBENCH’ section. It is also notable that there were thirteen analyses for which both the ‘mwTab’ and JSON formatted files contained errors.

It is also notable that for twenty-four (24) analysis IDs, both the ‘mwTab’ and JSON formatted files contained errors, which prevented the files from being parsed. For a full list of files containing parsing errors see Appendix A.

#### 2.2.3. Consistency Errors between ‘mwTab’ and JSON Formatted Files

After the analyses were filtered to remove those that were missing data files or contained processing errors, a total of 1655 analyses remained for which data files were present in both ‘mwTab’ and JSON formats. These analyses were then parsed into ‘~mwtab.mwtab.MWTabFile’ objects. We then compared the pair of objects representing a single analysis to determine if the parsed items were equivalent. Of the 1685 analyses, 1345 analyses had ‘mwTab’ and JSON formatted files that did not match (see Figure 1). Fourteen (14) analyses had ‘mwTab’ and JSON files which contained different section keys. This was commonly due to the extraneous inclusion of a blank ‘CHROMATOGRAPHY’ section within the ‘mwTab’ formatted data file. The remaining errors between formats can be broken down into; (1) errors in item sections (e.g., ‘PROJECT’, ‘STUDY’, ‘SUBJECT’, etc.), (2) errors in the ‘SUBJECT_SAMPLE_FACTORS’ section, and errors in data sections (e.g., ‘MS(NMR)_METABOLITES_DATA’, ‘NMR_BINNED_DATA’, ‘METABOLITES’, or ‘EXTENDED_MS(NMR)_METABOLITES_DATA’).

When validating the item sections, 198 analyses contained mismatched item keys within a given section, and 982 analyses had mismatched item values within a given section. Most of the analyses which had data files containing mismatched item keys within sections had additional item keys in the ‘mwTab’ formatted data file. Additionally, two (2) analyses had JSON files in which the ‘CHROMATOGRAPHY’ section was a list as opposed to the expected dictionary.

When validating the ‘SUBJECT_SAMPLE_FACTORS’ section of analyses, thirteen (13) analyses contained mismatched values in the section.

When validating the ‘MS(NMR)_METABOLITES_DATA’ and ‘NMR_BINNED_DATA’ sections (further referred to as ‘_DATA’ sections), 869 analyses had some inconsistency within the section. Four (4) of the analyses (AN001492, AN001493, AN001499, and AN002428) had errors due to their JSON files missing a ‘Data’ item representing the ‘mwTab’ files ‘MS_METABOLITES_DATA’ section. Five (5) analyses (AN000441, AN001960, AN002398, AN002403, and AN002404) contained mismatched ‘_DATA’ units items, all likely due to encoding differences in characters used in the unit’s value. The remaining analyses contained mismatched values within the individual data entries.

#### 2.2.4. Validation Errors in ‘mwTab’ Formatted Files

The downloaded ‘mwTab’ formatted files which could be parsed into ‘~mwtab.mwtab.MWTabFile’ objects were validated using the ‘~mwtab.validator.validate_file()’ method. Of the 1818 ‘mwTab’ files which could be parsed, 1539 of the files contained some validation error(s) (see Figure 2). In total, 1272 analysis files were found to have schema errors; 706 analysis files had null values in section items; and 391 data files had inconsistent sample IDs across their ‘SUBJECT_SAMPLE_FACTORS’ and ‘_DATA’ sections. Additionally, one (1) data file had inconsistent sample IDs across its ‘SUBJECT_SAMPLE_FACTORS’ and ‘EXTENDED_MS_METABOLITE_DATA’ sections. Thirty-seven (37) data files had missing ‘_DATA’ sections with no ‘MS_RESULTS_FILE’ item in their ‘MS’ section, effectively lacking any experimental data. Five (5) data files had null values in one or more fields within a sample data line in their ‘SUBJECT_SAMPLE_FACTORS’ section.

#### 2.2.5. Consistency Issues in ‘METABOLITES’ Section Metabolite Metadata Headings

MS and targeted NMR (files containing a ‘NMR_METABOLITES_DATA’ section) files from the downloaded analysis entries also had their metabolite metadata headings (field values) within their ‘METABOLITES’ section searched with Regular Expressions (RegExs) for fields which matched the standardized field names (i.e., pubchem_id, inchi_key, etc.). Of the 1818 downloaded ‘mwTab’ files, 1216 of the files contained field names which matched commonly used fields. The RegExs used and values matched can be seen in Table 4. While MW allows for users to specify these fields, the lack of consistency in field names across files presents a large issue hindering reuse of multiple studies in meta-analyses.

A few logical errors were also found with some of the user-specified field names during the matching process. One user-specified field name that matched the ‘retention_index’ standardized field name was ‘retention index (min)’. The retention index is a dimensionless measure as it is a normalization of a given compound retention time in relation to the retention times of two eluted standards. Additionally, it was found that some user-specified fields which matched the standardized ‘retention_time’ field contained unit denotations in the field name (e.g., ‘retention_time(min)’) while others did not. We suggest that depositors include a ‘retention_time_units’ heading for increased parsability.

### 2.3. Files Which Lack Data

MW serves as a repository for MS and NMR study data, and as a result, all files should contain either MS or NMR processed intensity data. If data are not explicitly included in the ‘MS_METABOLITE_DATA’ for NMR analyses, a ‘MS_RESULTS_FILE’ item should be included in the files ‘MS’. Currently, there is no equivalent ‘NMR_RESULTS_FILE’ item in MW’s format specification for the ‘NMR’ section. Further, 37 analyses from the downloaded ‘mwTab’ files contained no experimental data and failed to include a results file line. It is notable that result files from studies which do contain a ‘MS_RESULTS_FILE’ item can be downloaded through the MW File Transfer Protocol (FTP) server in the form of http://www.metabolomicsworkbench.org/Studies/”results_filename”.

## 3. Discussion

Since the first release of the ‘mwtab’ Python package, the MW repository and the ‘mwTab’ file format have seen a number of revisions and have expanded greatly. In order to keep up with MW, the ‘mwtab’ package has also seen a number of changes and bug fixes to match changes in both the ‘mwTab’ file format and MW’s web-based interfaces. The 1.0.1 release of the ‘mwtab’ package not only includes changes to update to the latest standards of MW, but also includes new functionality to improve programmatic access to MW. The package can now be used to work with MW’s JSON formatted analysis files. The package includes new validation functions that are useful in implementing or improving automatic data deposition pipelines and in facilitating multi-study meta-analyses. Further, the package includes new functionality to retrieve and interact with data from MW, which is not in its ‘mwTab’ file format. All improvements and expansions to the package are mirrored in both the API and the CLI. The extensive documentation for the ‘mwtab’ package has been expanded to document all new functionalities. The documentation includes a ‘User Guide’, ‘Tutorial’, and ‘API reference’ generated automatically from the source code and is still available at http://mwtab.readthedocs.io. The ‘mwtab’ package’s automated unit-tests have been expanded to test all new modules and functionality of the package and also generate test coverage reports.

We used the updated ‘mwtab’ package to check the metadata quality and data reusability of all available metabolomics datasets from the MW Data Repository. During our quality control and quality assessment (QC/QA) analysis, we found a large number of errors and consistency issues in data hosted by MW. When attempting to retrieve entries from the depository, fifty-three (53) analyses had blank entries for either their ‘mwTab’ or JSON formatted data file; 185 analyses contained gross formatting errors in either their ‘mwTab’ or JSON formatted data files (70 ‘mwTab’ and 139 JSON formatted files), which prevented the files from being parsed into ‘~mwtab.mwtab.MWTabFile’ objects; and 1345 of the 1655 analyses for which both ‘mwTab’ and JSON formatted data files could be downloaded contained inconsistencies across the file formats. A vast majority of ‘mwTab’ formatted files (1539 of 1818 files), which could be parsed into ‘~mwtab.mwtab.MWTabFile’ objects, contained validation errors in their content. Further, a large number of files (1216 of 1818 files) contained consistency errors in the naming of their metabolite metadata headings. Additionally, 37 analysis files lacked processed experimental data altogether.

We provided the complete QC/QA validation report to Metabolomics Workbench in early February 2021. At the time of acceptance of this paper, the vast majority of the missing and non-parsable entry files had been fixed by Metabolomics Workbench staff. Further, most of the consistency issues are being actively addressed.

There is a need for improving data and metadata quality and for the establishment of deposition standards [11,12,13,14]; however, we present the following suggestions that are narrowly focused on the repository format itself. Specifically, the prevalence of a large number of errors and consistency issues in data files from MW shows the need for improvements from the repository with respect to the implementation and maintenance of the ‘mwTab’ format. While MW has continued to update its file specifications, it has not updated many existing data files to the new standard. One way to address this could be to include named versions of the file specification and include the value as an item in the ‘METABOLOMICS WORKBENCH’ header sections. MW appears to include a ‘VERSION’ item line in the ‘METABOLOMICS WORKBENCH’ section, but it is unclear if the line specifies the version of the data file or the version of the ‘mwTab’ file specification. Additionally, the only version number present is “1” despite earlier data files not matching the updated file specification. A file specification version item line would allow for files hosted in a legacy specification to be easily identified from new up-to-date data files. Many public repositories include versioned file specification, such as the Protein Data Banks’ PDB file specification and Biological Magnetic Resonance Data Bank’s BMRB file specification, and these version specifications indicate major version, minor version, and patch or bug fix version of their respective format separated by a period [15,16]. Moreover, having both a data file version item line and a file format version item line is preferred so that both changes in content and format can be easily distinguished. We also recommend that MW implements methods to help standardize user-submitted headings. The methods should include common metabolite metadata headings matching in the ‘METABOLITES’ section along with whitespace stripping. These validation and consistency errors would hinder multi-study meta-analyses and large-scale computational analysis of the MW. Further, many of the consistency issues between ‘mwTab’ and JSON formatted data files appear to be caused by changes in character encodings. Therefore, we recommend adopting UTF-8 variable-width character encoding across both ‘mwTab’ and JSON formats. However, adopting UTF-8 encoding can be problematic for some legacy REST interface implementations.

In all fairness, many of the issues presented here are directly or indirectly due to a lack of diligence and effort from depositors. Moreover, the support and maintenance of a public scientific repository requires a community effort and should not be viewed as the sole responsibility of the repository itself, but what incentive do depositors have to do a good deposition, let alone a superior deposition, when a poor deposition is enough to satisfy the minimum data sharing requirements from journals and funding agencies? Therefore, we propose the idea of certification levels for depositions based on the quality and consistency of metadata provided. Since data quality is experiment dependent, the certification requirements would focus on metadata quality and indirectly data quality via the inclusion of data quality metrics. This idea has similarities to various curation metrics such as the UniProt Knowledgebase annotation score, which provides a heuristic score representing both the quality and quantity of the content supporting a UniProtKB entry or proteome [17,18]. However, there is a fundamental difference, since completeness of metadata to a certain high standard would be a requirement for the proposed certification. Furthermore, a certification system that includes a high-standard “gold” level would provide an incentive for depositors to do more than just the minimum required, since such certifications would help demonstrate research product quality and could be used in grant proposals and yearly progress reports. Since standard deposition would not require certification, depositors can still satisfy initial deposition requirements for publication while working towards refining depositions for certification. Further, the metabolomics repositories and the research community via scientific societies and standardization groups could work together to develop the requirements for deposition certification that would promote FAIRness and enable large-scale meta-analysis. Eventually, workshops could be developed for training depositors to reach certification.

## 4. Methods

### 4.1. Updates to the ‘mwTab’ Format

The ‘mwTab’ format specification was last updated on 5 February 2019. The updated ‘mwTab’ format specification is available on the MW website (mwTab file format specification. Available online: https://www.metabolomicsworkbench.org/data/mwTab_specification.pdf, accessed on 19 November 2020).

The ‘mwTab’ format has remained mostly the same with the exception of a few additions (see Figure 3). These additions to the ‘mwTab’ format include (1) the addition of DataTrack and Project IDs in the ‘METABOLOMICS WORKBENCH’ header line, (2) the requirement of a ‘RAW_FILE_NAME’ item in the ‘Additional sample data’ column (also now present as the ‘Raw file names and additional sample data’ column) of the ‘SUBJECT_SAMPLE_FACTORS’ section, (3) the expansion of key-value pairs in the ‘MS:MS_RESULTS_FILE’ lines in the ‘MS’ block, and (4) the allowance of a ‘NMR_METABOLITES_DATA’ block for targeted NMR studies. Many of these items are not explicitly stated as being updates to the ‘mwTab’ format, but as of November 2020 are present in a significant number of analyses.

Additionally, we have worked with the MW staff in developing one major improvement to the ‘mwTab’ file format and one convention in the format’s use, which have been accepted in depositions to MW. The major improvement to the ‘mwTab’ file format is the creation of the ‘EXTENDED_MS(NMR)_METABOLITE_DATA’ subsection in the ‘METABOLITES’ section, which allows the deposition of data that are both sample-specific and metabolite-specific and beyond the single intensity value allowed in the ‘MS(NMR)_METABOLITE_DATA’ block. Figure 4 illustrates this new addition to the ‘mwTab’ format, which includes both the metabolite identifier and the sample identifier along with any additional data and metadata field that are simultaneously sample-specific and metabolite-specific. This allows for the inclusion of various additional measurements such as peak-width and peak-height for NMR datasets and the retention time for chromatography-separated mass spectrometry. Moreover, various data quality metrics can now be included in a ‘mwTab’ entry such as an intensity standard deviation or an assignment confidence. The new format convention facilitates deposition of related subject and sample information as extra codified metadata in the ‘SUBJECT_SAMPLE_FACTORS’ block. This allows for the capture and representation of lineages of subjects and samples and the metadata associated with each subject and sample. For example (see Figure 5), a simple lineage could be cells extracted from a human subject for a cell culture experiment with downstream samples taken as aliquots for different analyses, but more complex subject-sample lineages could include multiple passages through a PDX mouse model. Further, the inclusion of these subject-sample lineages makes it much easier to link data across ‘mwTab’ entries from the same project representing separate analyses of different samples derived from the same group of subjects.

### 4.2. Metabolomics Workbench JSON File Format

When the ‘mwtab’ Python package was created, methods for converting MW’s ‘mwTab’ formatted files into JSON files were included. MW now allows for distribution of their data files in their own JSON format. The ‘mwtab’ Python package has been updated to mirror MW’s JSON format. The ‘~mwtab.mwtab.MWTabFile’ object is based upon an ordered dictionary structure. The dictionary structure of the object was changed to mirror the JSON format, allowing for ‘mwTab’ formatted files to be directly converted into their equivalent JSON format.

### 4.3. Overview of the Metabolomics Workbench REST Interface

The MW data repository provides multiple means of accessing the data stored. The Workbench provides a REST interface which allows for data to be accessed via HTTPS requests. REST URLs consist of three main parts; a context, an input specification, and an output specification (see Figure 6). The context specifies the type of data/resource intended to be accessed. The input specification describes the item to be requested and consists of two parts; an input item and an input value. The input specification is dependent upon the context requested. The output specification describes the output to be generated by the request and also consists of two parts; an output item and an output format. The full MW REST API specification is available on the MW website (Metabolomics Workbench REST URL-based API Specification. Available online: https://www.metabolomicsworkbench.org/tools/MWRestAPIv1.0.pdf, accessed on 19 November 2020).

### 4.4. Package Implementation

The ‘mwtab’ Python package previously consisted of seven modules: ‘mwtab.py’, ‘tokenizer.py’, ‘fileio.py’, ‘converter.py’, ‘mwschema.py’, ‘validator.py’, and ‘cli.py’. In the 1.0.1 release, changes were made to six of those modules (‘mwtab.py’, ‘tokenizer.py’, ‘fileio.py’, ‘mwschema.py’, ‘validator.py’, and ‘cli.py’), and two additional modules (‘mwrest.py’ and ‘mwextract.py’) were added to the package (see Figure 7). The updates to the original package modules either (1) mirrored updates made to the ‘mwTab’ file format, (2) greatly expanded the ‘mwTab’ format validation capabilities, or (3) incorporated new functionality provided by the ‘mwrest.py’ and ‘mwextract.py’ modules.

The ‘mwextract.py’ module implements a number of classes and functions for extracting metabolite data or metadata from ‘mwTab’ files (in both ‘mwTab’ and JSON format). The ‘~mwtab.mwextract.extract_metadata()’ method allows multiple ‘mwTab’ files to be searched and for a list of possible values to be generated from given section metadata keys. The ‘~mwtab.mwextract.extract_metabolites()’ method allows for multiple ‘mwTab’ files to be searched and for a list of metabolites to be generated from files containing a given set of metadata key-value pairs. The ‘~mwtab.mwextract.extract_metadata()’ method can be used to generate a list of metadata key-value pairs to use in the ‘~mwtab.mwextract.extract_metabolites()’ method.

The ‘mwrest.py’ module implements the ‘~mwtab.mwtab.GenericMWURL’ and ‘~mwtab.mwtab.MWRESTFile’ classes. The ‘~mwtab.mwrest.GenericMWURL’ class provides a programmatic representation of a URL constructed for use with MW’s REST API. The ‘~mwtab.mwrest.GenericMWURL’ class takes two parameters; (1) a string representing a base URL which directs users to MW’s REST API (defaults to https://www.metabolomicsworkbench.org/rest/) and (2) a dictionary of key-value pairs of REST parameters. The class validates that the given REST parameters are valid and then combines them with the base URL to create a specific URL which can be used to access MW’s REST interface. These changes do not alter the standard usage of the ‘~mwatb.fileio.read_files()’ method, but enable these methods to take the ‘url’ parameter from a ‘~mwtab.mwrest.GenericMWURL‘ object and retrieve ‘mwTab’ formatted files. Behind the scenes, these methods perform HTTPS requests to retrieve the described data and/or metadata from the MW REST interface, provided the url specifies a valid entry which can be represented as an ‘mwTab’ formatted entry. Further, the functionality of the package has been increased beyond dealing with ‘mwTab’ formattable data. The ‘~mwtab.mwrest.MWRESTFile’ class mirrors the ‘~mwtab.mwtab.MWTabFile’ class in the ‘mwtab.py’ module, but serves as a representation of files which cannot be represented in an ‘mwTab’ format. There is now a ‘~mwtab.fileio.read_mwrest()’ function for downloading data which cannot be represented in an ‘mwTab’ file (ie. csv, json, and plain text data).

The ‘validator.py’ module has seen a large number of improvements and now includes a number of new methods for validating additional sections of ‘mwTab’ files. The module previously contained methods for validating sample and factor ids and the section schemas of ‘mwTab’ files. The ‘validator.validate_file()’ method existed for these functions. The method has now been improved to include gathering of all existing validation errors and searches for an expanded number of errors. Methods now exist for validating the metabolites and metadata headings in the ‘METABOLITES’ block and for validating the data in the ‘MS(NMR)_METABOLITE_DATA’ or ‘NMR_BINNED_DATA’ blocks. All of these specific validation methods are called by the ‘~mwatb.validator.validate_file()’ method.

The ‘cli.py’, ‘fileio.py’, ‘mwschema.py’, and ‘mwtab.py’ modules have all been updated to match the changes to the ‘mwTab’ file format and include the new functionality provided from the new ‘mwrest.py’ and ‘mwextract.py’ modules. For the full documentation, see the Read the Docs page at https://mwtab.readthedocs.io/ [19].

As of 1 January 2020, Python 2 reached its end-of-life and will no longer be receiving new bug reports, fixes, or changes from the Python Software Foundation [8]. As a result, the 1.0.1 release of the ‘mwtab’ Python package was not developed with Python 2 support, and no future updates are planned to include Python 2 support. The 0.1 releases of the ‘mwtab’ package which does support Python 2 will, however, remain available through both GitHub and PyPI.

### 4.5. Evaluation of the Metabolomics Workbench Repository

The ‘mwtab’ package functionality was again evaluated on every available ‘mwTab’ and JSON formatted file available from MW (as of 19 November 2020). A total of 1891 analysis IDs were available, and the downloading of both ‘mwTab’ and JSON format data files was attempted through the MW REST interface.

The files from MW were evaluated for a number of formatting and consistency errors/issues via a set of tested assertions. First, the successful download of each analysis file through MW’s REST API was tested in both ‘mwTab’ and JSON format. Second, the successful parsing of each downloaded file into MWTabFile objects was tested. Third, the consistency between parsed data from ‘mwTab’ and JSON formats was tested. Fourth, each parsed ‘mwTab’ formatted file was validated using the validation methods in the updated ‘mwtab’ Python package. Only ‘mwTab’ formatted data files where validated with the validation methods used would likely give fairly redundant results across both ‘mwTab’ and JSON formats. Lastly, the analysis file which contained ‘METABOLITES’ sections had its metabolite metadata headings parsed with RegExs which were used to detect commonly used fields. This was used to detect consistency issues in field names across analyses, which could hinder cross-study analysis.

The method used to download all analysis files became the ‘download all’ CLI function. The method uses the ‘~mwtab.mwrest.analysis_ids()’ function to retrieve a list of available analysis IDs from MW. The method then retrieves and saves a data file for each of the available analysis IDs. We called the method twice: once to retrieve ‘mwTab’ formatted files (--to-format = txt) and once again to retrieve JSON formatted files (--to-format = json).

We then attempted to parse each downloaded file into a ‘~mwtab.mwtab.MWTabFile’ object.

Next, we used the ‘~mwtab.validator.validate_file()’ method to validate the contents of each resulting ‘~mwtab.mwtab.MWTabFile’ object. The method consists of four main sub-validation functions. The first function validates the consistency of sample ID and sample factors across the file. This function ensures that there are no blank sample ids or factors and that a subset of the sample ids from the ‘SUBJECT_SAMPLE_FACTORS’ block is present in the ‘_DATA’ blocks. The second function validates the consistency of metabolite names across the ‘MS(NMR)_METABOLITE_DATA’ and ‘METABOLITES’ blocks in MS files and targeted NMR files. The function ensures that there are no blank metabolite names and that the metabolites are present in both blocks of the file. The third function validates that no non-numeric values are present in MS processed experimental data. The fourth and final function validates each file against the expected ‘mwTab’ file format (‘mwTab’ schema). The function ensures that each required section is present with the appropriate metadata items.

MS analysis files contain a ‘METABOLITES’ section which consists of a table of metabolites and metabolite metadata headings (i.e., fields). Commonly used fields include but are not limited to: Human Metabolome Database ID (hmdb_id), International Chemical Identifier (inchi_key), KEGG ID (kegg_id), mass over charge value (moverz), quantified mass over charge value (moverz_quant), PubChem ID (pubchem_id), retention index (retention_index), and retention time (retention_time) [20,21,22,23]. MW’s deposition policy allows users to specify the names of these fields, which makes it difficult to accurately parse these field names due to spelling inconsistencies. Therefore, we created RegExs to detect common aberrations so that a list of standardized field names can be used.

### 4.6. Updates to the ‘mwtab’ Package Documentation

Updates to the codebase were consistently documented using the Sphinx Python documentation style [24]. All additional documentation (‘User Guide’, ‘Tutorial’, ‘API Reference’, etc.) was also updated and is available at https://mwtab.readthedocs.io/.

The updated package documentation includes improvements to the docs/tutorial.ipynb file which contains basic information about the ‘mwTab’ file format as well as mwtab package usage details and examples. Additional Jupyter notebooks (api_examples.ipynb, cli_examples.ipynb, and diabetes_search.ipynb) have been included in the FigShare repository which provides a range of examples for using the application programming interface, the command line interface, as well as details on how to use the added mwextract module functionality to search directories of ‘mwTab’ formatted files for analyses associated with specific diseases (with the given example being diabetes).

## Figures and Tables

**Figure 1 metabolites-11-00163-f001:**
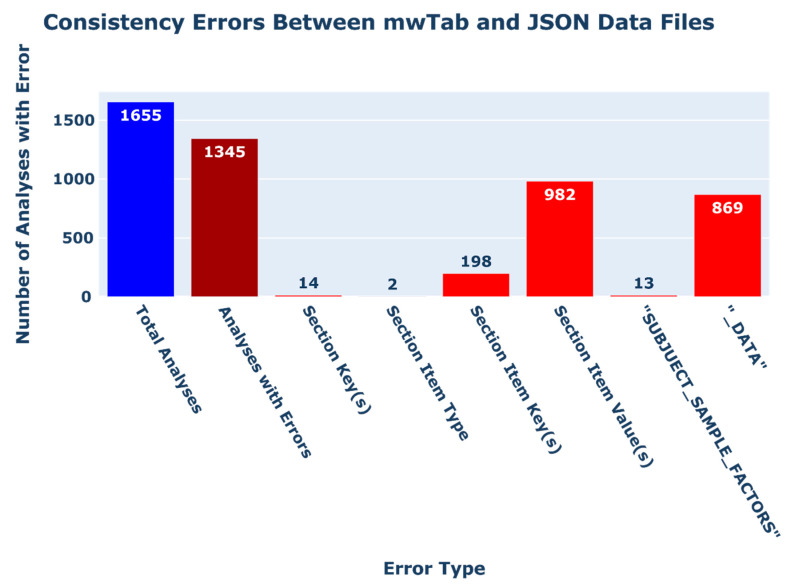
Bar chart depicting the count of consistency errors between ‘mwTab’ and JSON formatted data files. The total number of analyses for which both ‘mwTab’ and JSON formatted files were available upon download and could be parsed into ‘~mwtab.mwtab.MWTabFile’ objects is shown in blue on the left. The total number analyses with inconsistencies between the two available data file formats is shown in dark red second from the left. The remaining red bars show the number of analyses which possess each given inconsistency type.

**Figure 2 metabolites-11-00163-f002:**
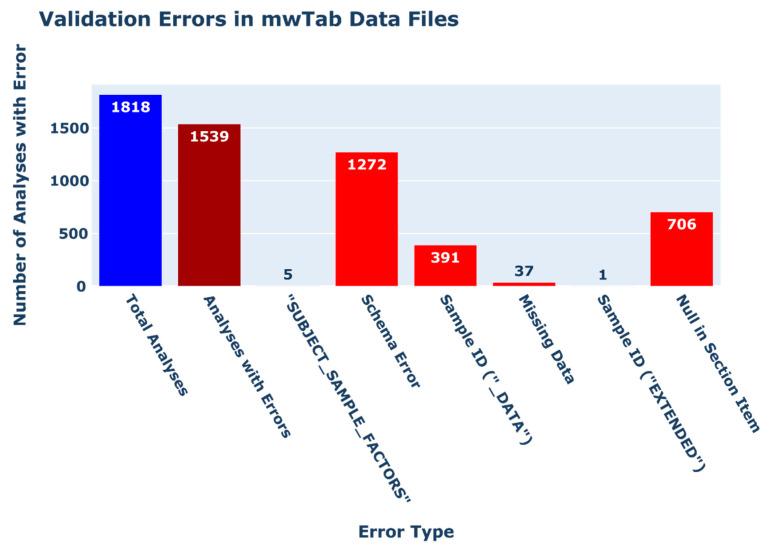
Bar chart depicting the count of validation errors in ‘mwTab’ formatted data files. The total number of analyses for which ‘mwTab’ files were available upon download and could be parsed into ‘~mwtab.mwtab.MWTabFile’ objects is shown in blue on the left. The total number of analyses with validation errors is shown in dark red second from the left. The remaining red bars show the number of analyses which possess each given error type.

**Figure 3 metabolites-11-00163-f003:**
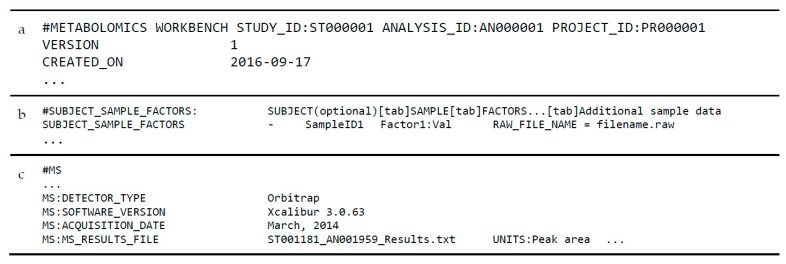
Overview of the updated ‘mwTab’ format: (**a**) Updated header block which now includes ‘PROJECT_ID’ key-value pair; (**b**) Uploaded ‘SUBJECT_SAMPLE_FACTORS’ block format with the additional ‘RAW_FILE_NAME’ field in the ‘Additional sample factors’ column; and (**c**) Updated ‘MS’ block format with additional key-value pairs appended to ‘MS:MS_RESULTS_FILE’ line.

**Figure 4 metabolites-11-00163-f004:**
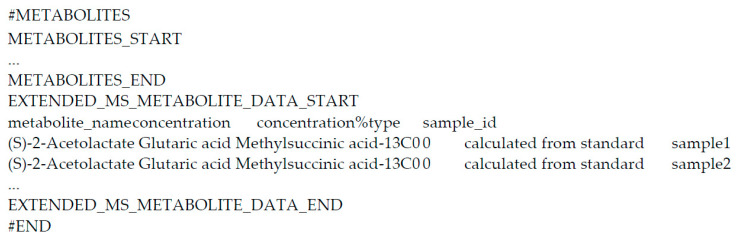
Overview of an ‘EXTEND_MS_METABOLITE_DATA’ section in the ‘METABOLITES’ block of an example ‘mwTab’ file.

**Figure 5 metabolites-11-00163-f005:**
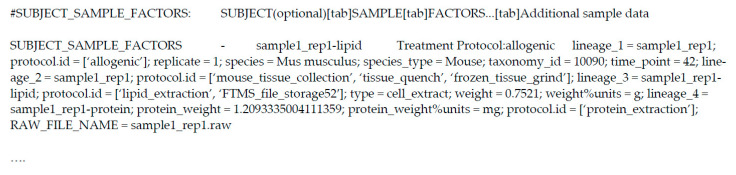
Overview of a ‘SUBJECT_SAMPLE_FACTORS’ block of an example ‘mwTab’ file. Additional sample-specific data and metadata can be included in this block as key-value pairs separated by semicolons (as highlighted). While their inclusion may appear cluttered, these key-value pairs are computer parsable and therefore interoperable.

**Figure 6 metabolites-11-00163-f006:**
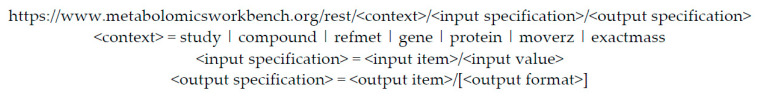
Pattern for crafting HTTPS requestable URLs to interface with the Metabolomics Workbench REST interface.

**Figure 7 metabolites-11-00163-f007:**
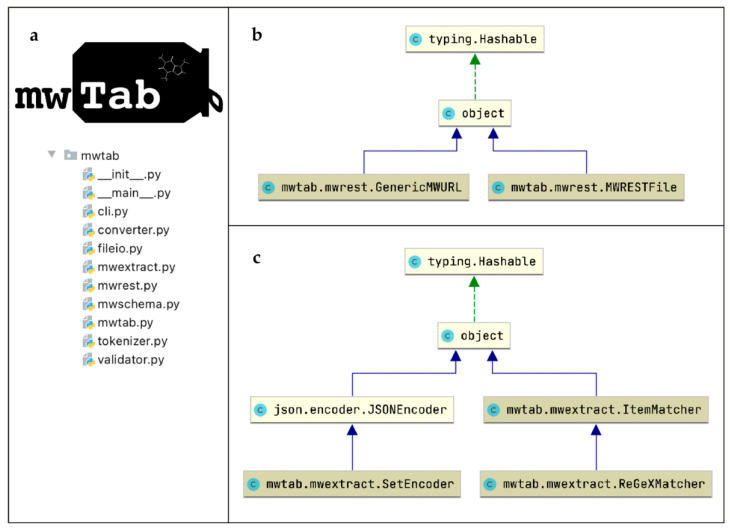
Organization of the ‘mwtab’ Python package represented with Unified Modeling Language (UML) diagrams: (**a**) UML package diagram of the ‘mwtab’ Python library; (**b**) UML class diagram of the ‘fileio.py’ module; (**c**) UML class diagram of the ‘mwtab.py’ module; d. UML class diagram of the ‘converter.py’ module.

**Table 1 metabolites-11-00163-t001:** Common patterns for using the ‘mwtab’ as a library.

Usage	Example
Reading ‘mwTab’ Files From REST	# create first REST URL mwt_rest_url = mwtab.GenericMWURL({ “context”: “study”, “input_item”: “analysis_id”, “input_value”: “AN000002, “output_item”: “mwtab”, “output_format”: “txt”}).url # create second REST URL another_mwt_rest_url = next(mwtab.generate_mwtab_urls(“AN000003”)) # create a generator to call REST URLS and create MWTabFile objects mwt_generator = mwtab.read_files(mwt_rest_url, another_mwt_rest_url)
Extracting Metadata	# make a generator to create MWTabFile objects mwtabfile = next(fileio.read_files(“path-to-mwtabfile-dir”)) # extract metadata extracted_values = mwextract.extract_metadata(mwtabfile, “LAST_NAME”)
Extracting Metabolites	# make a generator to create MWTabFile objects mwtabfile_gen = fileio.read_files(“path-to-mwtabfile-dir”) # create matcher object matcher = mwextract.generate_matchers([(“PR:LAST_NAME”, “Smith”)]) # extract metabolites data metabolite_dict = mwextract.extract_metabolites(mwtabfile_gen, matcher)

**Table 2 metabolites-11-00163-t002:** Command patterns for using the ‘mwtab’ as a Command-Line Interface.

Command	Description	Example
download	Download files through the Metabolomics Workbench REST API	$ mwtab download all $ mwtab download 1 $ mwtab download AN000001 $ mwtab download ST000001 --context=study \ --input-item=study_id --output-item=mwtab \ --output-format=json $ mwtab download C20H34O11 --context=compound \ --input-item=formula --output-item=all --output-format=txt
extract	Extract data or metadata from file(s)	$ mwtab extract metadata AN000001.txt./LAST_NAME \ CHROMATOGRAPHY_TYPE --to-format=csv $ mwtab extract metabolites file_dir/./PR:LAST_NAME \ Smith CH:CHROMATOGRAPHY_TYPE “Reversed phase” \ --to-format=json

**Table 3 metabolites-11-00163-t003:** List of analysis IDs and the format of files that could not be downloaded through Metabolomics Workbench’s REST API.

File Format	Analysis ID
‘mwTab’	AN002380, AN002381, and AN002384
JSON	AN000255, AN000404, AN000405, AN000410, AN000415, AN000436, AN000439, AN000444, AN000446, AN000450, AN000663, AN000665, AN000667, AN000871, AN001856, AN002131, AN002132, AN002133, AN002134, AN002135, AN002136, AN002137, AN002138, AN002141, AN002142, AN002145, AN002147, AN002148, AN002149, AN002150, AN002151, AN002152, AN002153, AN002154, AN002157, AN002158, AN002159, AN002160, AN002161, AN002162, AN002163, AN002164, AN002165, AN002166, AN002167, AN002168, AN002169, AN002170, AN002171, and AN002314

**Table 4 metabolites-11-00163-t004:** List of common metabolite metadata headings from ‘METABOLITES’ blocks of MS and targeted NMR analyses, Regular Expressions (RegExs) used to match field names, and examples of similar/matching fields present in analysis files. See Appendix A for a full list of matched field names.

Common Field Name	RegEx Pattern(s)	Example Matched Field Names
hmdb_id	r”(?i)[\s|\S]{,}(HMDB)” r”(?i)(Human Metabolome D)[\S]{,}”	HMDB ID (*representative) HMDB (*Representative ID) HMDB_ID … Total 14 Fields
inchi_key	r”(?i)(inchi)[\S]{,}”	Inchi_Key InChIKey InchiKey … Total 10 Fields
kegg_id	r”(?i)(kegg)$” r”(?i)(kegg)(\s|_)(i)”	KEGG KEGG I Kegg ID … Total 6 Fields
moverz	r”(?i)(m/z)”	m/z M/Z m/z rounded
moverz_quant	r”(?i)(moverz)(\s|_)(quant)” r”(?i)(quan)[\S]{,}(\s|_)(m)[\S]{,}(z)”	Quantified m/z quantitated mz Moverz Quant … Total 10 Fields
other_id	r”(?i)(other)(\s|_)(id)$”	Other ID Other_ID
pubchem_id	r”(?i)(pubchem)[\S]{,}”	PubChem CID Pubchem ID PubChem … Total 9 Fields
retention_index	r”(?i)(ri)$” r”(?i)(ret)[\s|\S]{,}(index)”	retention time index ri Retention index … Total 9 Fields
retention_time	r”(?i)(r)[\s|\S]{,}(time)[\S]{,}”	retention_times retention time index Retention Time … Total 20 Fields

## Data Availability

The data presented in this study, supplemental materials, scripts used, and version of the updated mwtab Python package used are openly available in FigShare at https://doi.org/10.6084/m9.figshare.12094104.

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
