# Peer review of "The mwtab Python Library for RESTful Access and Enhanced Quality Control, Deposition, and Curation of the Metabolomics Workbench Data Repository"

_metabolites, 2021, doi:10.3390/metabo11030163_

Round 1
Reviewer 1 Report
The manuscript presents mwtab, a python library to assist and improve the quality and acceptability of the Metabolomics Workbench (MW) repository. While reading the manuscript, I felt disappointed about the quality of MW contributions. Fortunately, mwtab library will help to improve the quality of the contributions in the MW repository. The package is a great addition for metabolomics science (and disciplines beyond) and aims to follow the FAIR data principles of findability, accessibility, interoperability, and reusability. The manuscript is well-written, English is flawless and the text is condensed as it should be. My favorite part was the lines 245-268, which goes beyond the description of the technical aspects of mwtab. I would advise the authors to create Jupyter Notebooks with mwtab workflows to increase the visibility of their work. The manuscript is a timely addition to the Journal and I suggest to be accepted as it is.
Author Response
Reviewer 1:
The manuscript presents mwtab, a python library to assist and improve the quality and acceptability of the Metabolomics Workbench (MW) repository. While reading the manuscript, I felt disappointed about the quality of MW contributions. Fortunately, mwtab library will help to improve the quality of the contributions in the MW repository. The package is a great addition for metabolomics science (and disciplines beyond) and aims to follow the FAIR data principles of findability, accessibility, interoperability, and reusability. The manuscript is well-written, English is flawless and the text is condensed as it should be. My favorite part was the lines 245-268, which goes beyond the description of the technical aspects of mwtab. I would advise the authors to create Jupyter Notebooks with mwtab workflows to increase the visibility of their work. The manuscript is a timely addition to the Journal and I suggest to be accepted as it is.
Response:
We thank the reviewer for recognizing the effort we have put into this package and manuscript. Also, when we submitted this manuscript for review, we also sent the validation report to Metabolomics Workbench staff and are working with them to address all of the issues highlighted in the manuscript. Already, almost all of the missing files and gross parsing issues have been fixed by Metabolomics Workbench. We have added this information to the manuscript as follows:
“We have provided the complete QC/QA validation report to Metabolomics Workbench in early February 2021. At the time of acceptance of this paper, the vast majority of the missing and non-parsable entry files have been fixed by Metabolomics Workbench staff. Also, most of the consistency issues are being actively addressed.”
Also, as per your request, we have added three Jupyter notebooks to the FigShare repo demonstrating a range of examples for the application programming interface, the command line interface, and a disease-specific use-case. This is in addition to the Jupyter notebook already in the FigShare repo that generates the validation report.
Reviewer 2 Report
This is interesting topics and upgraded for the previous python package. The findings and the analysis of this manuscript is very significant for data scientist community, however, I do not know how the user can benefit from it. I recommended authors give an example ad a notebook which cover how user who is looking for a data set from the workbench can use it and advance their research. For example, if the user is interested in diabetes metabolomics dataset, so how can this package help him to answer the question including the quality control step and the metabolites consensus analysis.
Author Response
Reviewer 2:
This is interesting topics and upgraded for the previous python package. The findings and the analysis of this manuscript is very significant for data scientist community, however, I do not know how the user can benefit from it. I recommended authors give an example ad a notebook which cover how user who is looking for a data set from the workbench can use it and advance their research. For example, if the user is interested in diabetes metabolomics dataset, so how can this package help him to answer the question including the quality control step and the metabolites consensus analysis.
Response:
We thank the reviewer for recognizing the significance of our work. To address your request, we have added a Jupyter notebook and its conversion to HTML demonstrating a diabetes use-case for finding relevant MW entries and extracting metabolites for evaluating coverage. Also, we have included two other Jupyter notebooks demonstrating example utilization of the package’s application programming interface and command line interface.